# Crosstalk between Oxidative Stress and Aging in Neurodegeneration Disorders

**DOI:** 10.3390/cells12050753

**Published:** 2023-02-27

**Authors:** Rehab F. Abdelhamid, Seiichi Nagano

**Affiliations:** 1Department of Neurology, Osaka University Graduate School of Medicine, Suita 565-0871, Osaka, Japan; 2Department of Neurotherapeutics, Osaka University Graduate School of Medicine, Suita 565-0871, Osaka, Japan

**Keywords:** oxidative stress, free radical, aging, neurodegenerative diseases

## Abstract

The world population is aging rapidly, and increasing lifespan exacerbates the burden of age-related health issues. On the other hand, premature aging has begun to be a problem, with increasing numbers of younger people suffering aging-related symptoms. Advanced aging is caused by a combination of factors: lifestyle, diet, external and internal factors, as well as oxidative stress (OS). Although OS is the most researched aging factor, it is also the least understood. OS is important not only in relation to aging but also due to its strong impact on neurodegenerative diseases such as amyotrophic lateral sclerosis (ALS), frontotemporal dementia (FTD), Alzheimer’s disease (AD), and Parkinson’s disease (PD). In this review, we will discuss the aging process in relation to OS, the function of OS in neurodegenerative disorders, and prospective therapeutics capable of relieving neurodegenerative symptoms associated with the pro-oxidative condition.

## 1. Introduction

Healthy aging is characterized by a gradual breakdown of physiological systems leading to a reduction in cognitive functions and brain health, but the timing and extent of this decline vary among older people. Oxidative stress (OS) is a crucial factor in the aging process that can cause direct damage to the brain’s cellular architecture, causing neurodegenerative disease.

Aging is the main lead factor for many diseases, including cancer, metabolic, cardiac and neurodegenerative diseases. Aging is linked to a loss of homeostasis, involving degradation of structural components, reduced cellular maintenance and a decrease in overall physiological function/metabolism. There are two major aging theories: the free radical theory, which postulates chronological accumulation of defects in gene expression and environmental damage. According to the free radical theory of aging by Denham Harman, 1956 [1], alterations in normal metabolic and mitochondrial function are induced by production of free radicals, which cause damage, aging, and associated aging illnesses.

The second and more recent theory is the mitochondrial theory of aging by J. Miquel and colleagues in 1980 [2]. This theory posits that the loss of balance between free radical production and repair mechanisms is responsible for aging.

The redox network is also essential in antioxidant defense. Mitochondria serve a crucial function by turning stored energy into adenosine triphosphate (ATP) through oxidative phosphorylation and phospholipid synthesis, buffering calcium, and coordinating programmed cell death [3]. Free radicals such as reactive oxygen species (ROS), reactive nitrogen species (RNS), and reactive sulfur species (RSS), which are present in all cells, but which are restricted by the antioxidant systems that neutralize them, are primarily produced by oxidative phosphorylation [4]. Free radicals are produced naturally and modulate cellular processes such as inflammation, cell survival, and stress responses, as well as numerous illnesses such as cardiovascular problems, muscular dysfunction, allergies, and malignancies [5].

Free radicals are most likely formed due to processes involving molecular oxygen that are catalyzed in the cell by oxidative enzymes and in connective tissues by trace metals such as iron, cobalt, and manganese.

The primary distinction between healthy aging and accelerated aging is the balance between free radical elevation and the body’s ability to guard and fight against RNS and ROS (Figure 1).

In the event of loss of homeostasis between free radical production and detoxification, ROS production may overpower antioxidant defenses, resulting in a noxious state known as OS and general degradation of normal cellular functions. This has been documented in numerous clinical studies involving mitochondrial malfunction and aging [6].

The brain is especially prone to oxidative damage due to its high oxygen consumption, limited antioxidant defenses, and high concentration of polyunsaturated fatty acids that are easily oxidized [7].

Lipid peroxides (LPO) are highly reactive molecules that include malondialdehyde (MDA), 4-hydroxy-2-nonenal (HNE), acrolein, isoprostanes (IsoPs), and neuroprostanes (NeuroPs). They can disrupt proteins and DNA structure and functions [8,9,10]. Increased MDA, IsoPs, and HNE have been observed in brain tissues of Tg2576 Alzheimer’s disease (AD) model mice and post-mortem AD brains [11,12]. Oxidative damage to DNA results in formation of oxidized base adducts, including 8-hydroxyguanosine (8-OHG) and 5,6-diamino-5-formamidopyridine in brains of mild cognitive impairment (MCI) patients. Despite the brain’s great potential for ROS formation, its defense system against OS remains restricted and diminishes with age, owing to low amounts of endogenous antioxidants such as catalase (CAT), glutathione (GSH), glutathione peroxidase (GPx), and vitamin E, compared to other tissues, such as liver [13,14]. The limited regenerative ability of postmitotic neurons renders OS more damaging to the brain, where the damage accumulates over time, compared to other organs [15].

## 2. Other Sources of Free Radicals Contribute to OS

Even though mitochondrial dysfunction is the primary source of free radicals, there are several other sources, such as genetic (endogenous), environmental, and lifestyle causes (exogenous) [16].

### 2.1. Environmental Factors: Radiation/UV Rays and Pollution

Environmental factors, including smoking, UV radiation, heavy metal ions, ozone, allergens, drugs or toxins, pollutants, and pesticides, may all contribute to elevated ROS production in cells [17,18]. Ionizing radiation transforms hydroxyl radicals, superoxides, and organic radicals into organic hydroperoxides and hydrogen peroxide. Subsequently, peroxides react with metal ions, particularly Fe and Cu through redox reactions, with further oxidative activity at the cellular level. Several studies have demonstrated that exposure of fibroblasts to alpha particles leads to an intracellular increase in oxygen and an acceleration of peroxide formation [19,20]. Ultraviolet radiation (UVA) induces oxidative processes by stimulating riboflavin, porphyrins, and NADPH-oxidase, resulting in synthesis of 8-oxo-guanine and a drop in intracellular GSH levels, with a return to normal after exposure cessation [21]. Heavy metals are crucial to the formation of free radicals [22].

Nickel, arsenic, iron, copper, cadmium and lead can produce free radicals by Fenton or Haber-Weiss reactions [23,24], as well as through direct reactions between metal ions and cellular constituents with similar effects, such as generation of thiol-type radicals. In brain tissue, lead causes lipid peroxidation and raises GPx concentrations. By attaching to sulfhydryl groups, arsenic generates peroxides, superoxides, and nitric oxide, and inhibits antioxidant enzymes such as glutathione-S-transferase (GST), GPx, and glutathione reductase (GR) [25]. Free radicals produced by these reactions can cause substitutions of DNA bases, including guanine for cytosine, guanine for thymine, and cytosine for thymine [26]. Even in healthy individuals, ozone exposure can impair lung function by increasing inflammatory infiltration in the respiratory epithelium [27].

### 2.2. Lifestyle Related Factors

Lifestyle factors such as smoking, drinking alcohol, diet, exercise, and frequency of exercise all contribute to OS [28]. Some research has shown that ROS are present at the skeletal muscle level and help to control how muscle works. Muscle fibers always produce small quantities of reactive oxygen radicals, which are elevated by muscle contraction [29]. These oxygen radicals have many direct and indirect effects on muscle activity (contractility, excitability, metabolism, and calcium homeostasis) and are involved in skeletal muscle fatigue during hard exercise [30].

Long, exhausting exercise and overcoming limits as a phase of overtraining cause a massive response to OS. Endogenous antioxidant status is improved by moderate exercise, low-intensity training, and training for a long time.

ROS are an essential part of how cells talk to each other and how antioxidant genes are turned on and off. Nuclear factor kappa B and mitogen-activated protein kinase are up-regulated by physical activity [31,32]. These activate gene expression of various enzymes and proteins that maintain oxidative/antioxidant intracellular homeostasis [33].

Physical exercise is the main non-drug therapy for treating chronic diseases, especially heart diseases, and lifestyle changes [34]. Relevant studies [35] have shown that autophagy, a process that breaks down and recycles cellular organs and nutrients, is important for cardiovascular benefits of training.

Cigarette smoke has oxidants, free radicals, and organic components such as nitric oxide and nitric superoxide. These cause neutrophils and macrophages to gather in lung tissues, increasing the production of oxidants in the area [36,37].

### 2.3. Genetic Factors

Enzymes that are responsible for fighting damage caused by OS and these enzymes and pathways are prone to free radical damage [38,39,40,41,42]. Oligodendrocytes are susceptible to oxidative damage due to their function in maintenance and creation of myelin and their limited repair mechanisms, suggesting that white matter may be more susceptible to oxidative activity than gray matter. Antioxidant defense enzymes in the brain, such as superoxide dismutase (SOD), CAT, GPx, and GST are essential for neutralizing toxic byproducts of oxidative phosphorylation. Allelic variants of polymorphisms encoding these antioxidants are associated with anomalies in SOD, CAT, GPx, and GST activity in the central nervous system [38,43,44].

## 3. Mechanistic Evolution of Neurodegenerative Diseases Caused by OS

Neurodegenerative disease is an irreversible condition in which neuronal function declines over time, leading to neuronal death. The incidence of neurodegenerative disease is increasing every year, especially in countries with aging populations. Common neurodegenerative diseases include Alzheimer’s (AD) and Parkinson’s (PD) diseases. Mitochondria are the primary source of ROS and are the main cause of neurodegenerative diseases (Figure 2).

Mitochondrial synthesis of ATP through oxidative phosphorylation, has the substantial disadvantage of producing unpaired electrons [45,46]. The electron transport chain comprises five multiprotein complexes that mediate interaction of these electrons with oxygen, resulting in ROS hydrogen peroxide (H_2_O_2_), superoxide anions (O_2^−^_), and hydroxyl radicals (•OH) [14,47]. Electron transfer from NADH to ubiquinone is catalyzed by mitochondrial complex I (reduced nicotinamide adenine dinucleotide [NADH] coenzyme Q reductase) (coenzyme Q). Complex II also provides electrons to ubiquinone (succinate dehydrogenase). Electrons from reduced ubiquinone are donated to complex III (cytochrome bc1), and subsequently, to cytochrome c (CytC). Complex IV (CytC oxidase) is involved in interactions between molecular oxygen and electrons extracted from CytC, resulting in water production [48]. Complexes I, II, and III are most often linked to premature electron leakage to oxygen and are the primary source of ROS generation [49].

Additionally, elevated ROS levels trigger formation of additional reactive species, such as RNS, when O_2_ reacts with other molecules, such as nitric oxide, to generate peroxynitrite (ONOO^−^). Furthermore, in addition to ROS and RNS, mitochondria generate RSSs, which are very reactive. Free oxygen radicals gradually damage proteins, lipids, and nucleic acids, resulting in inefficient or aberrant cellular functions, inflammation, and cell death [50].

Mitochondrial internal components and mitochondrial DNA (mtDNA) in particular, are very vulnerable to OS-induced damage, resulting in impaired mitochondrial bioenergetics, increased ROS generation, and OS [51]. Enzymatic and non-enzymatic mechanisms safeguard antioxidant systems. SOD, CAT, GPx, GR, and thioredoxin (TRX) are the major enzymes involved in catalytic ROS elimination. Non-enzymatic complexes include vitamins A, C, and E, GSH, and proteins such as albumin and ceruloplasmin [52,53].

## 4. Major Degenerative Disorders Caused by OS

### 4.1. OS in Amyotrophic Lateral Sclerosis (ALS)

Despite substantial investigation, the genesis of amyotrophic lateral sclerosis (ALS) is still not fully clear. Ninety percent of ALS cases are sporadic and appear to lack a genetic foundation, whereas 10% of patients have familial ALS, mostly an autosomal dominant [54]. ALS has been linked to various occupational and environmental variables, such as exposure to various chemicals, metals, and pesticides, electromagnetic fields (EMFs), and lifestyle choices, including smoking and excessive exercise [55,56,57,58].

A recognized but poorly understood pathogenic ALS feature is abnormally high free radical levels and inadequate antioxidation. Undoubtedly, OS is essential for motor neuron death, but we do not know exactly when oxidative damage occurs [59]. OS biomarkers have been identified in brain tissue of ALS patients, cerebrospinal fluid (CSF), blood, and urine [60]. It is challenging to track OS biomarkers over a long period due to the short lifespans of ALS patients. Further, because of OS’s random start and the present lack of tools to forecast its progression, it is difficult to be certain whether OS is a cause of ALS-associated neurodegeneration or a result of other underlying etiologic variables [57]. Investigations using a murine ALS model have revealed altered mitochondrial structures and nuclear factor erythroid 2-related factor 2 (Nrf2) pathway activation during early ALS stages, indicating OS involvement in the disease’s early stages. OS damage is typically caused by stimulating formation of intracellular antioxidant molecules [61,62].

However, these studies used the murine mutant SOD1 ALS model, and SOD1 mutations only account for 20% of human familial ALS cases. Twenty individuals with sporadic ALS had significantly higher levels of lipid peroxidation and lower levels of the antioxidant enzymes CAT, GR, GSH, and glucose-6-phosphate dehydrogenase (G6PD) in their erythrocytes [63]. Progression of alterations parallel the pathophysiology of ALS, supporting OS participation in ALS development. Additionally, the aforementioned environmental and endoplasmic reticulum stress factors work together to promote pro-oxidative states that may ultimately harm motor neurons [64].

The intermembrane space (IMS) of mitochondria contains a protein known as Coiled-Coil-Helix-Coiled-Coil-Helix Domain-Containing Protein 10 (CHCHD10), which has no known function [65]. However, it is believed to be involved in maintaining mitochondrial crista morphology and proper oxidative phosphorylation. In particular, in multiprotein complexes I, II, III, and IV, overexpression of mutant CHCHD10 harboring an allele linked to ALS alters mitochondrial structure and impairs electron transport chain function [66,67]. In addition, fibroblasts from an ALS patient with a CHCHD10 mutation displayed mitochondrial ultrastructural damage and mitochondrial network fragmentation [66,67].

### 4.2. OS in Alzheimer’s Disease (AD)

The most common neurodegenerative condition, AD, is characterized by a steady decline in behavior, cognition, and functioning that profoundly affect day-to-day activities. AD brains have higher protein, DNA, and lipid oxidation rates, as well as redox-active metals [51].

Pathological indicators of AD include extracellular senile plaques (SP) and intracellular neurofibrillary tangles (NFT). Protein aggregation is sporadic, and its molecular mechanism is poorly understood. Several investigations have revealed that AD brains display elevated OS, which is crucial to disease development [68]. Amyloid fibril precursors are neurotoxic owing to OS generated by toxic conformer of amyloid oligomers and additional neurotoxic effects such as neuronal membrane rupture, microglia and astrocyte activation, and Ca^2+^ dyshomeostasis [69,70,71].

Tau protein aggregation is another hallmark of AD although in most cases, it is of late onset. Several reports have indicated that neurotoxicity caused by beta amyloid alteration is the main trigger of tau alteration to form tangles. NFTs in the early stages of AD are intracellular deposits; however, these progress to extracellular deposits in later stages. NFTs are composed of paired helical filaments (PHFs) in which the major component is tau protein [72,73,74,75,76].

When tau proteins are hyperphosphorylated, they self-assemble into NFTs and may be detected in neurons [77,78]. According to recent data, amyloid deposition occurs 15–20 years before dementia manifests itself, and tau pathology appears thereafter [79,80,81,82,83].

Using multiphoton imaging, researchers identified a clear link between free radical generation and amyloid plaques in AD mouse models and human AD brain tissues, where fluorogenic free radical markers decreased following administration of a synthetic antioxidant [84].

### 4.3. OS in Parkinson’s Disease (PD)

Unlike AD, PD is clinically recognized by four cardinal motor symptoms: bradykinesia, stiffness, resting tremors, and trouble with posture and walking [83,84,85]. In PD brains, the substantia nigra pars compacta and to a lesser extent, the globus pallidus, putamen, and caudate nucleus, exhibit selective dopaminergic neuronal loss. The nigrostriatal pathway’s degenerating neurons release less of the neurotransmitter dopamine [86]. Lewy bodies are clusters of aberrant proteins seen in neurons of individuals with Parkinson’s disease. They are components of α-synuclein (α-syn) protein, broadly disseminated in the neurological system, but activities of which are not well understood [87]. α-syn fibrillation creates clumps that take up significant space within cells and ultimately result in neuronal death [88]. As in AD, pathophysiological pathways underlying PD are strongly related to OS [89]. Evidence of OS involvement may be seen in the substantia nigra of PD patients, where oxidized lipids, proteins, and DNA can be found [90,91]. Additionally, the monoamine oxidase (MAO) that breaks down dopamine creates hydrogen peroxide, whereas dopamine on its own produces superoxide anion and reactive quinones. These reactive substances cause other nearby neurons, as well as dopaminergic neurons, to become cytotoxic [92,93].

### 4.4. OS in Multiple Sclerosis (MS)

Multiple sclerosis (MS) is a multifactorial autoimmune disease of the central nervous system (CNS), characterized by chronic inflammation, demyelination, and axon and neuron loss. Depending on the location of the demyelinating lesions, MS patients can develop almost any neurological sign or symptom, including motor, sensory, and cognitive impairment [94,95].

OS is heavily involved in several MS pathological hallmarks such as myelin destruction, axonal degeneration, and inflammation [96]. In MS as in other neurodegenerative diseases OS triggers activation of autophagy and microglia as well as of the neuroimmune system. Neurons, astrocytes, and oligodendrocytes produce chemicals that bind to microglial receptors in healthy settings, suppressing their activated states [97,98,99,100]. When certain molecules (such as myelin CD47) are expressed less, microglial activation is increased, which may cause myelin debris to be phagocytosed and to provide neurotrophic factors [101,102]. Microglial physiological functions are transformed into harmful inflammatory insults by long-term damage, systemic inflammation, proinflammatory cytokine release, and ROS signaling [103]. Together, our results imply that during the onset and development of experimental autoimmune encephalomyelitis (EAE) and MS lesions, activated microglia and macrophages are directing tissue damage through their oxidative burst. Simultaneous activation of a sophisticated antioxidant response is insufficient to stop the apoptotic and degenerative processes, however.

Due to a combination of circumstances, including high levels of iron and polyunsaturated fatty acids, high iron requirements and mitochondrial activity, and restricted cell regeneration, the CNS is particularly susceptible to OS. Oxidative damage also affects the immune response that is developing in the periphery and controls MS illness outside of the CNS. First, by lowering its electrical resistance and thus changing its permeability, elevated ROS levels harm the brain endothelium [94,104,105,106].

## 5. Significance of Non-Coding RNAs in OS

OS readily damages RNA due to its single-stranded structure and high concentration close to mitochondria, where most intracellular ROS are generated [107,108]. An overabundance of ROS can cause chemical modification or even separation of RNA bases and breakage of RNA strands [91,109]. Oxidatively damaged RNA accumulates in cells, leading to decreased protein synthesis, erroneous protein generation, and ultimately cell death [110,111].

Non-coding RNAs (ncRNAs), which do not encode proteins, make up the majority of RNAs in human cells [112,113]. As a category of ncRNAs, regulatory ncRNAs, including long ncRNAs (lncRNAs) and small ncRNAs, which includes microRNAs (miRNAs), circular RNAs (circRNAs), and PIWI-interacting RNAs (piRNAs), are involved in regulating gene expression [114]. Compared to mRNAs, these regulatory ncRNAs persist relatively longer. Hence, oxidative damage can cause prolonged effects. ncRNAs of several types have been linked to various neurogenerative diseases [115,116,117,118]. However, the direct relationship between ncRNA oxidation and neuronal diseases is still unclear, except in PD, in which the interaction between OS and regulatory ncRNAs has been well studied [119]. In contrast, regulatory small ncRNAs such as miRNAs and lncRNAs help to regulate ROS production [118]. Their interactions are involved in the pathophysiology of PD [120].

RNA oxidation is not only a hallmark of PD but also a crucial first step in the development of the disorder [112,121]. 8-oxo-7, 8-dihydroguanosine (8-oxoG) is one of many oxidative marks on RNA, and it is possibly the most common and well studied [122]. Guanine’s vulnerability to free radicals means that it can form this base adduct, leading to mismatched bases [109].

Neuronal apoptosis and autophagy are influenced by lncRNAs; thus, α-syn accumulation and degradation that restricts their function would have deleterious effects on cellular homeostasis [123]. For instance, free radical damage to miRNAs can cause them to misidentify their target mRNAs, which can increase expression of specific proteins [124]. A reduction in α-syn expression was mediated by two miRNAs in an experiment performed by Je and Kim [123]. High levels of α-syn and subsequent development of PD resulted from stress-induced oxidative loss of translational inhibition by these two miRNAs.

Furthermore, Chen et al. found that OS triggered modification of circRNA N6-methyladenosine (m6A). m6A-modified circRNAs can modulate expression of stress response genes (UBC and PPP2CA), which may constitute the mechanistic basis for OS-induced neurodegenerative disorders [125].

The regulatory role of miRNAs in OS is closely linked to α-syn, which is responsible for inducing OS. Both microRNA-141-3p and microRNA-9-5p target the 3′ UTR of SIRT1 mRNA. Since SIRT1 inhibits formation of α-syn aggregates, knockdown of microRNA-141-3p and microRNA-9-5p may alleviate OS and boost the viability of in vitro PD model [126,127].

SOD, CAT, and GPx are responsible for detoxifying oxidants and repairing oxidative damage. MicroRNA-137 and microRNA-494-3p aggravate OS by reducing SOD and GPx activity in PD rats treated with these miRNAs. MicroRNA-335 suppresses expression of FTH1, thereby promoting release of Fe^2+^ ions and generation of free radicals [128]. Downregulation of microRNA-410 expression in PD is associated with elevated ROS production [129].

Regulatory lncRNAs and OS are hallmarks of PD. In PD, mitochondrial dysfunction is linked to ROS overproduction. α-syn aggregate formation can exacerbate OS by decreasing complex I activity or by activating microglia. Upregulation of beta-amyloid-cleaving enzyme-antisense (BACE1-AS), a lncRNA, was also associated with increasing levels of α-syn in PD [130]. In addition, the lncRNA microRNA-17-92a-1 cluster host gene (MIR17HG) promotes α-syn expression. MicroRNA-153-5p cannot inhibit α-syn expression because MIR17HG acts as a sponge for microRNA-153-5p [131]. Inhibition of autophagy by GSK3 promotes α-syn buildup and therefore aggregation. MicroRNA-15b-5p suppressed GSK3 expression, but binding of SNHG1 to microRNA-15b-5p rescued GSK3 expression [132]. Cellular effects of oxidatively damaged RNA and the mechanism by which regulatory ncRNAs affect OS are just beginning to be explored.

## 6. Methods to Measure OS in Neurodegenerative Diseases

Although there are peripheral indicators for OS, it is difficult to detect in the human brain in vivo. The following measures have been used to quantify brain OS (Table 1).

### 6.1. Peripheral Blood OS

Several surrogate OS or antioxidant activity indicators, including circulating lipid peroxides, GSH, and vitamins C and E, have been examined in peripheral blood [89,133,134,135].

AD patients display reduced peripheral vitamin A, C, and E levels [136] and SOD and GPx activity [137]. Lower plasma GSH levels are associated with more severe cognitive impairment in AD patients [138]. PD patients have inconsistently altered erythrocyte SOD activity [139,140]. Increased SOD activity may protect against OS damage.

### 6.2. Magnetic Resonance Spectroscopy (MRS)

GSH is the single antioxidant assessed by ^1^H (MRS) [149]. GSH content in the human brain is lower than N-acetyl aspartate, creatine, and choline, making evaluation difficult. With MRS, it is challenging to discern between GSH and glutamate resonance peaks [141,150]. GSH-CH_2_ cysteine’s protons resonate at 2.93 and 2.97 ppm, overlapping with those of creatine (3.03 ppm) and aspartate (2.82 ppm) [151]. To detect GSH levels in the brain, spectral editing methods such as MEscher-GArwood-PRESS (MEGA-PRESS) [142] are needed to boost GSH signals so as to gather reliable nuclear signals. MEGA-PRESS combined with a 180° editing pulse in the original PRESS pulse sequence [142] may distinguish GSH-cysteine signals from other signals, notably creatine signals in the brain [89]. Recent research found that GSH levels are lower in AD patients’ hippocampus and frontal cortices [143,144].

The concentration of vitamin C (ascorbic acid) in the human brain is around 1.0 mM, which may be detected by MRS [152]. However, measuring vitamin C using ^1^H MRS is problematic owing to the similarities between the resonances of vitamin C (3.73, 4.01, and 4.50 ppm) and glutamate (3.75 ppm) [151]. MEGA-PRESS editing might potentially aid in measuring vitamin C levels in the human brain. Several earlier research [153] used ^1^H-MRS with the MEGA-PRESS to evaluate vitamin C levels in the human brain.

### 6.3. Electron Paramagnetic Resonance (EPR) Spectroscopy

Electromagnetic radiation absorption often occurs in the microwave region of the electromagnetic spectrum [154]. As a result, it is influenced by paramagnetic species that are present in a magnetic field. However, owing to short radical half-lives, compared to the EPR time scale, EPR spectroscopy cannot detect them directly [155]. To compensate for this, a stable chemical is frequently used to capture radicals to make them observable [156].

EPR has been implement in detecting neurodegenerative diseases as in AD [145] and as real time OS marker for post stroke patients [146].

Despite years of research in EPR spectroscopy, one of the main reasons why it has not been widely employed is presumably poor sensitivity, particularly at the levels of free radicals usually encountered in biological systems. More research is required before using EPR in human clinical trials.

### 6.4. Positron Emission Tomography (PET)

The intracellular over-reactive state can be measured using a radiotracer for positron emission tomography (PET) [147]. [^62^Cu] diacetyl-bis (N4-methylthiosemicarbazone) ([^62^Cu] ATSM) is a radiotracer widely used in PET. Evaluation of striatal OS in patients with PD using [^62^Cu]ATSM PET was previously confirmed [147]. In a recent study, this tracer was utilized to visualize localized OS in PD patients that was mostly caused by mitochondrial malfunction. In this investigation, deposition of [^62^Cu] ATSM was detected in the striatum of PD and ALS patients [147,148], indicating a localized over-reductive conditions caused by mitochondrial malfunction.

## 7. Approaches to Slow Aging by Fighting OS

The human body has several lines of defense against OS. Multiple antioxidant defense pathways are involved in the brain’s ROS detoxification. The metalloproteins SOD, CAT, and GPx constitute the first line of antioxidant defense against ROS. Age-related accumulation of highly reactive polyunsaturated fatty acids, iron, and ROS is exacerbated by suboptimal antioxidant levels.

Apart from endogenous antioxidant defense mechanisms in aging brain and given the importance of OS in neurodegenerative diseases, manipulating ROS levels may be a viable therapeutic approach to slow neurodegeneration and reduce related symptoms.

Avoiding OS-related causes is at the top of preventative approaches. Diet, exercise, lack of sleep, sedentary behavior and circadian rhythm abnormalities are all crucial components in regulating healthy aging. The gut flora is hypersensitive to extrinsic variables linked to an unhealthy lifestyle [157,158]. Preclinical studies showed that a high-fat diet (HFD) in mice changes the gut microbiome [159]. Pro-inflammatory bacteria such as Clostridium, Eubacterium, and Roseburia are positively correlated with hyperglycemic fluctuations in the brain [160]. The need for gut microbiome therapies is growing, and this area of research is continually developing. With impressive results, dietary and probiotic supplementation has been investigated as a possible therapeutic strategy for age-related disorders via changes in gut microbiota. Recent research has shown that exercise may alter gut microbiota, adding to the potential advantages of this strategy for treating disorders related to aging [161].

Nutraceuticals or natural compounds exist in food have been extensively studied worldwide due to their neuroprotective effects in vivo and in vitro, attributed to the antioxidative properties [162]. Nutraceuticals that show metal chelation ability and anti-inflammatory properties such as curcumin [163,164], green tea polyphenols [164], resveratrol [165] and vitamin E [166,167] are examples of natural antioxidants from foods and other sources that show promise as therapeutic agents for OS-related neurodegenerative diseases.

## 8. Conclusions and Concluding Remarks

Aging is not a disease that needs to be treated but a natural process; however, premature or unhealthy aging still needs more study to understand mechanisms and leading causes. There is a controversial view that OS leads to a short life span and is linked to age-related disease and quality of life. Numerous studies have shown a significant elevation in OS and free radical concentration as a common cause of aging and accompanying neurodegenerative diseases. However, in nature, comparative studies indicate long lives of some animals despite high levels of free radicals in their tissues. This means that there are many unknown mechanisms to cope with OS. Therefore, a healthy lifestyle to avoid factors that elevate OS, as well as increased intake of natural antioxidants, can protect against OS and prevent neurodegenerative diseases before their onset.

## Figures and Tables

**Figure 1 cells-12-00753-f001:**
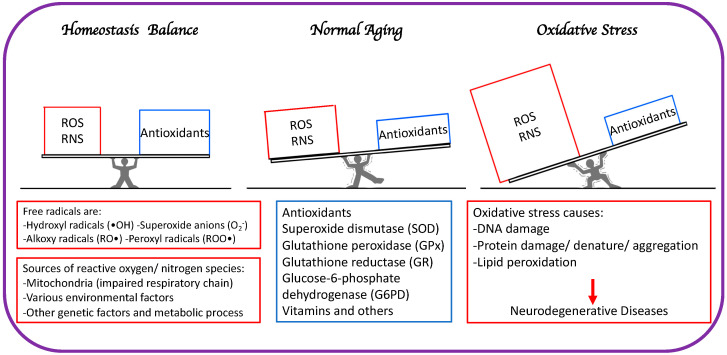
The primary distinction between healthy aging and accelerated aging is the balance between free radical generation (RNS and ROS) and antioxidation to minimize oxidative stress.

**Figure 2 cells-12-00753-f002:**
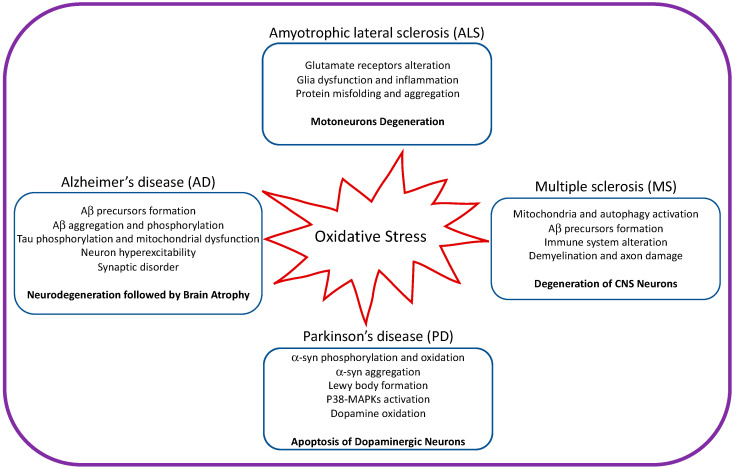
Illustration showing the impact of oxidative stress on amyotrophic lateral sclerosis, Alzheimer’s disease, Parkinson’s disease, and multiple sclerosis.

**Table 1 cells-12-00753-t001:** Overview of the methods and biomarkers that are used to detect oxidative stress in neurodegenerative diseases.

Biomarker	Sample	Method	Disease Detected	References
LPO, GSH, GPx, SOD and vitamins C and E.	Peripheral blood	Biochemical analysis (Blood screening)	AD, PD	[133,134,135,136,137,138,139,140]
GSH and Vitamin C. using ^1^H MRS	Brain	^1^H Magnetic Resonance Spectroscopy (MRS)	AD, MCI	[141,142,143,144]
Intracellular over-reductive state using redox sensitive probes	Brain	Electron Paramagnetic Resonance (EPR)	AD, Depressive disorders in post-stroke patients	[145,146]
Intracellular over-reductive state using Radiotracer.	Brain	Positron Emission Tomography(PET)	PD, ALS	[147,148]

MCI: Mild cognitive impairment; LPO: lipid peroxidation; GPx: glutathione peroxidase; GSH: glutathione; SOD: superoxide dismutase.

## Data Availability

Not applicable.

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
