# Peer review of "Crosstalk between Oxidative Stress and Aging in Neurodegeneration Disorders"

_cells, 2023, doi:10.3390/cells12050753_

Round 1
Reviewer 1 Report
In this review, Abdelhamid and Nagano describe the role of oxidative stress in aging and neurodegenerative diseases. However, I have a few concerns that should be addressed:
- The antioxidant system should be introduced adequately.
- The role of oxidative stress in aging is poorly described.
- Section 2 (sources of free radicals led to oxidative stress) should be further discussed.
- Additional literature in the topic should be included to enrich the manuscript.
Minor concerns:
- Line 138: The sentence should be corrected “Genetic factors are the group of enzymes”.
- Line 147: references are missing.
Reviewer 2 Report
The authors have discussed a nice topic regarding neurodegenerative disorders and oxidative stress. The article is quite interesting, and I do agree that reactive oxygen species play a vital role in neurodegeneration, oxidative stress regulation, and so on. However, it is full of writing without sufficient illustrations, so readers can easily lose attention. I would recommend you to write as simply as possible with some tables and figures. Overall, the merit of the manuscript is good and follows a logical way. Still, there are some major modifications that the authors need to be addressed in their revised version.
-
Please write the symbols in proper ways like OH- not OH- in Figure 1, Section 3. It should be checked throughout the manuscript for all the chemical structures.
-
Please give cartoon-like drawings for this section as much as possible. As it is full of writing, the readers can easily be distracted.
-
AD is characterized by Ab plaque and tau aggregation. This section shows some evidence of Ab plaque but not tau aggregation. Please include one paragraph for this. There are a few articles on tauopathy and OS published. The authors can review some of them.
-
Could you please provide a tabular format of the biomarkers with references in Section 6.
-
I believe the title should be changed as the story is all about reactive oxygen species with neurodegeneration, oxidative stress, aging, etc. So, it would be more fit if the authors bring the term Crosstalk among oxidative stress, aging, neurodegenerative disorders or something like this.
-
Could you please make a tabular format or graphical format stating the regulatory effects of OS on neurodegenerative disorders or disease names? That would be easier to understand the roles and significance of OS..
Round 2
Reviewer 2 Report
Dear author,
Thanks for correcting all the concerns and now it is ready for publication.
Congratulations....